# Saline gargle collection method is comparable to nasopharyngeal/oropharyngeal swabbing for the molecular detection and sequencing of SARS-CoV-2 in Botswana

Kwana Lechiile,[1,2,3] Sikhulile Moyo,[1,2,4,5] Mai-Lei Woo Kinshella,[6] Wonderful T. Choga,[2] Leabaneng Tawe,[3] Jonathan Strysko,[3] Gofaone Bagatiseng,[2] Iryna Kayda,[6] Kedumetse Seru,[2] Boitumelo J. L. Zuze,[2] Patience Motshosi,[2] Mosepele Mosepele,[2,7] Irene Gobe,[1] Simani Gaseitsiwe,[1,2] Margaret Mokomane,[1] David M. Goldfarb[3,6]

**ABSTRACT** The coronavirus disease 2019 pandemic has highlighted the importance and challenges of the sample collection component of the diagnostic cycle. Although combined nasopharyngeal and oropharyngeal swabs (NOS) have historically been the gold standard of sampling, the saline gargle (SG) sampling method has been evaluated and implemented in multiple jurisdictions for respiratory pathogen detection. It has proven to be user-acceptable to patients, simple to collect, and highly sensitive to severe acute respiratory syndrome coronavirus 2 (SARS-CoV-2) detection by molecular methods when compared to swabs. We performed a prospective cross-sectional study to evaluate the SG collection method against the NOS collection method for molecular detection and next-generation sequencing (NGS) of SARS-CoV-2 in Botswana. Paired SG and NOS samples were collected and underwent nucleic acid extraction prior to molecular detection. The SG had an overall sensitivity of 81.3% (95% CI: 68.8%%–96.0%), while the NOS had an overall sensitivity of 96.9% (95% CI: 84.3–99.4). Paired samples with a mean crossing threshold value of <35 also underwent NGS. SG specimens had a median genome coverage of 94.7% (interquartile range [IQR] 87.0%–99.2%) and NOS specimens had a median genome coverage of 99.6% (IQR 90.0%–99.6%). Bioinformatics analysis showed the 15 successfully matched pairs belong to clades BA.1 and BA.2 indicative of the Omicron variant. Further analysis at the nucleotide level showed a mean similarity of 99.998% ± 0.00465% between NOS and SG. This method has the potential to overcome the challenges that come with swab-based sampling for SARS-CoV-2 testing and may be an alternative in testing for other viral pathogens.

**IMPORTANCE** During the coronavirus disease 2019 (COVID-19) pandemic, a major challenge has been inadequate sampling for detection of severe acute respiratory syndrome coronavirus 2 (SARS-CoV-2). Pediatric patients posed additional challenges with sample collection, and they and others are also at risk of rare complications from swab collection. Saline gargle (SG) sampling method has been evaluated and introduced as an alternative to swab collection in several jurisdictions. Our study affirms the acceptable performance of the saline gargle method for the molecular detection of SARS-CoV-2 and also establishes that SG samples do not pose an obstacle for genomic sequencing of SARS-CoV-2. The SG method may be a reliable alternative for SARS-CoV-2 detection and next-generation sequencing, facilitating COVID-19 surveillance efforts in resource-constraint settings.

**KEYWORDS** diagnostics, SARS-CoV-2, gargle

Address correspondence to Kwana Lechiile, lechiilek@hotmail.com.

The authors declare no conflict of interest.

See the funding table on p. 9.

The prompt and accurate detection of severe acute respiratory syndrome coronavirus 2 (SARS-CoV-2) is essential in the identification of positive cases, implementation of interventions to control community transmission, evaluation and monitoring of coronavirus disease 2019 (COVID-19) vaccine effectiveness, and large-scale surveillance for new SARS-CoV-2 variants (1). Laboratory molecular diagnostic testing that comprised nucleic acid extraction followed by reverse transcription polymerase chain reaction (RT-PCR) has been classified as the gold standard testing method for the detection of SARS-CoV-2 (2). One of the main challenges in SARS-CoV-2 testing during the pandemic has been in the collection of an adequate diagnostic sample (3). Botswana implemented the collection of combined nasopharyngeal and oropharyngeal swab (NOS) samples for testing for SARS-CoV-2 infection (4), utilizing guidance provided by the World Health Organization (WHO) (5). This sampling method can be challenging, and it is often associated with discomfort for patients (6) and severe complications in rare instances (7, 8). NOS sampling requires the services of a healthcare worker (HCW) for proper collection, the use of personal protective equipment, and is resource-intensive for resource-limited settings (3). It can pose an added challenge for the pediatric populations when NOS manufactured for children and adolescents are not easily accessible, and those manufactured for adults are often too large for sufficient sample collection. There is a need to increase access to non-invasive sampling for accurate SARS-CoV-2 testing that is accommodating for all populations.

The saline gargle (SG) collection method for SARS-CoV-2 detection has been evaluated and implemented in multiple jurisdictions, including in Canada where it was offered as an alternative to swab collection for the school-aged and youth population to address these testing challenges and shortages in material resources (9, 10). Several studies in Europe, Asia, and North America have also reported SG collection as an effective alternative to NOS collection for COVID-19 testing (1, 3, 10–13). The SG sample has proven to be simple to collect, user-acceptable for patients, amenable for laboratory processing, and highly sensitive in detecting SARS-CoV-2 by molecular methods, when compared to swabs (3, 9). SG samples may also be amenable for use in ongoing SARS-CoV-2 sequencing. Next-generation sequencing (NGS) has been instrumental for continued surveillance throughout the COVID-19 pandemic, and local implementation of routine SARS-CoV-2 sequencing resulted in Botswana being the first to detect and alert the world of the highly transmissible Omicron variant (14). To date, only a few reported studies have assessed the performance of SG collection for SARS-CoV-2 in a low- and middle-income country and its capacity for genomic sequencing.

We performed a prospective cross-sectional study to evaluate the performance of the SG collection method against the NOS collection method in the molecular detection and next-generation sequencing of SARS-CoV-2 in Botswana.

## RESULTS

### Patient demographics

A total of 127 participants who provided matched samples were enrolled in the study across the two recruitment periods (Table 1). There were 66 females (52.0%) included. The median age of participants was 14 years (interquartile range [IQR] 11–31.75 years), and 78 (61.4%) of the participants were <18 years of age. Among the participants, 84 (66.1%) were symptomatic while 25 (19.7%) were asymptomatic individuals attending the clinic for contact tracing. The remaining 18 (14.2%) were tested as part of travel requirements or routine checks for employment. Table S1 displays an overall breakdown of patient demographics for all participants, including cycle threshold (Ct) value PCR results for their NOS and SG samples.

### qPCR testing

The results of qPCR testing are shown in Tables 2 and 3. There was a total of 31 positive NOS samples, 26 of which were also positive on the SG samples. The SG samples had

**TABLE 1** Summary of study participant demographics

| Characteristic | *N* (%) or Median [IQR] |
|---|---|
| Total patients | 127 |
| Age (years) | 14 [11–31.75] |
| Sex | |
| Male | 61 (48.0) |
| Female | 66 (52.0) |
| Clinical presentation | |
| Symptomatic | 84 (66.1) |
| Asymptomatic | 43 (33.9) |
| Molecular testing assay | |
| DaAn Gene | 92 (72.4) |
| GeneXpert | 35 (27.6) |

six false negatives, and there was one sample which was positive on the SG sample and negative on the NOS sample. The SG had an overall sensitivity of 81.3% (95% CI: 68.8%-96.0%) while the NOS had an overall sensitivity of 96.9% (95% CI: 84.3–99.4).

## Sequencing and bioinformatics analysis

A total of 25 matched SG and NOS samples that were PCR-positive for SARS-CoV-2 with mean Ct values <35 were taken for NGS and bioinformatics analysis ( Table S2). While the SG specimen had a median genome coverage of 94.7% (IQR 87.0%–99.2%), the NOS specimen had a median genome coverage of 99.6% (IQR 90.0%–99.6%) ( Fig. S1). The overlapping IQRs suggest that the genome coverages in the two sample types may be comparable; however, statistical significance was detected ($P < 0.001$), indicating that NOS sampling has significantly higher coverage when compared to SG sampling.

Of the 50 samples that underwent NGS, 41 had successful nucleotide sequences generated, of which 20 were SG samples and 21 were NOS samples. Of the 41 successful sequences, 32 were paired samples. Nucleotide sequences with matched pairs whose NOS samples had at least 80% genome coverage underwent phylogenetic analysis, and Fig. 1 shows the results of the molecular phylogenetic tree for 15 matched pairs (30 samples). NextClade analysis showed the matched pairs belong to clades BA.1 and BA.2 indicative of the Omicron variant.

When plotted with SARS-CoV-2 strains circulating in Botswana at that time, the samples were shown to have similar strains as the referenced accession numbers and to belong to the two Omicron clades. Further analysis at the nucleotide level showed a mean similarity of 99.998% ± 0.00465% between NOS and SG.

Genetic mutation analysis showed that the majority of sequenced samples had mutations of high affinity to angiotensin-converting enzyme-2 receptors on the spike gene ( Table S3). One sample (KJL025_SG) showed no mutations, and we observed missing regions that were not sequenced. An example is displayed for one of the successful paired samples for KJL011 where the NOS sample (EPL_ISL_15158766) and its matched SG sample (EPL_ISL_15158765) had the same mutations in the spike genes (Fig. 2). The mutations in the specimen types shared similar variants, such as G142D on the NTD region of the spike gene, and S375F, N440K, and Q493R in the receptor-binding domain (RBD) region, which also indicates the circulation of the Omicron variant.

**TABLE 2** Performance of saline gargle (SG) vs reference standard in SARS-CoV-2 detection

| Saline gargle | Reference standard | | | Value, % (95% CI) |
|---|---|---|---|---|
| | Positive | Negative | Total | Sensitivity |
| Positive | 26 | 0 | 26 | 81.3 (68.8–96.0) |
| Negative | 6 | 95 | 101 | |
| Total | | 95 | 127 | |

**TABLE 3** Performance of nasopharyngeal/oropharyngeal swab (NOS) vs reference standard in SARS-CoV-2 detection

| Nasopharyngeal/oropharyngeal swabs | Reference standard | | | Value, % (95% CI) |
|---|---|---|---|---|
| | Positive | Negative | Total | Sensitivity |
| Positive | 31 | 0 | 31 | 96.9 (84.3–99.4) |
| Negative | 1 | 95 | 96 | |
| Total | | 95 | 127 | |

## DISCUSSION

In order for efforts against the COVID-19 pandemic to be effective, the implementation of a reliable diagnostic cycle is essential. A target product profile group at the WHO recommends that an acceptable sensitivity for a SARS-CoV-2 assay should be set at 80% (15). We found that the SG collection method had reached this sensitivity (81.3%) when compared with the highest standard of sample collection (combined NOS). These findings are consistent with observed findings from several similarly conducted studies in other regions with 93.8%–98% sensitivity of SG against nasopharyngeal swabs in Canada (1, 3, 16), 100% sensitivity of gargle lavage against nasopharyngeal swabs in Germany (12), and 86.2% sensitivity of SG against nasal and throat swabs in India (17). We observed higher Ct values among positive samples in the SG when compared to the swab samples (median 28.3 vs 21.8), which has also been reported by other studies (1, 18).

A possible explanation for the lower sensitivity observed in our study compared to other studies may be attributed to the collection of combined nasopharyngeal and oropharyngeal swabs as the comparator method. NOS yields the greatest quality sample and the highest sensitivity, even when compared to nasopharyngeal swabs alone (19, 20). It may also be due to sample dilution, as the swabs were eluted in 3 mL of viral transport media, while the SG method involved 5 mL of saline solution. The larger volume in the SG method may have resulted in the further dilution of the patient's sample prior to testing.

Given the challenges of the sample collection due to patient discomfort (21), risk of potentially severe complications (22), inconsistent supply of NOS, and the need for skilled HCWs, the SG sampling method has the ability to address these issues. Firsthand experiences from our study discovered that participants favored SG over NOS as they considered SG collection to be less painful. User acceptability of SG collection has been formally evaluated in at least two prior studies (references 3 and 23), and both found that these sample types had significantly higher acceptability ratings. The high patient acceptability of saline gargle is a unique benefit of this sampling method, which may be an optimal solution for those undergoing frequent testing or patients, such as children, who have difficulty with invasive swab collections.

The SG collection kit used in this study comprises a 5 mL 0.9% saline solution, a funnel to aid the collection of samples, and a sterile collection tube. In comparison, the swab collection kit in the Botswana setting comprises a nasopharyngeal swab, an oropharyngeal swab, and a collection tube with 3 mL viral transport media (4). The consumables of the SG kit are widely available and have lower chances of stockout. This swab-independent, alternative sample type may minimize the challenges of inconsistent supply of sample collection resources in Botswana, particularly for pediatric patients whose specimen collection kits are not easily accessible.

Healthcare workers have the greatest exposure to ill patients in testing facilities and therefore have the highest risk of contracting infection. For this reason, they are required to wear appropriate personal protective equipment (PPE) when conducting a NOS sampling method. They must also be skilled and have undergone sufficient training in order to obtain adequate samples without inflicting unnecessary harm or discomfort on the patient. The SG method has shown that a skilled HCW for sample collection is not necessary. Kinshella et al. found that SARS-CoV-2 could be detected with high sensitivity using self-collected SG, even when the self-collection was unobserved by a HCW (1). By

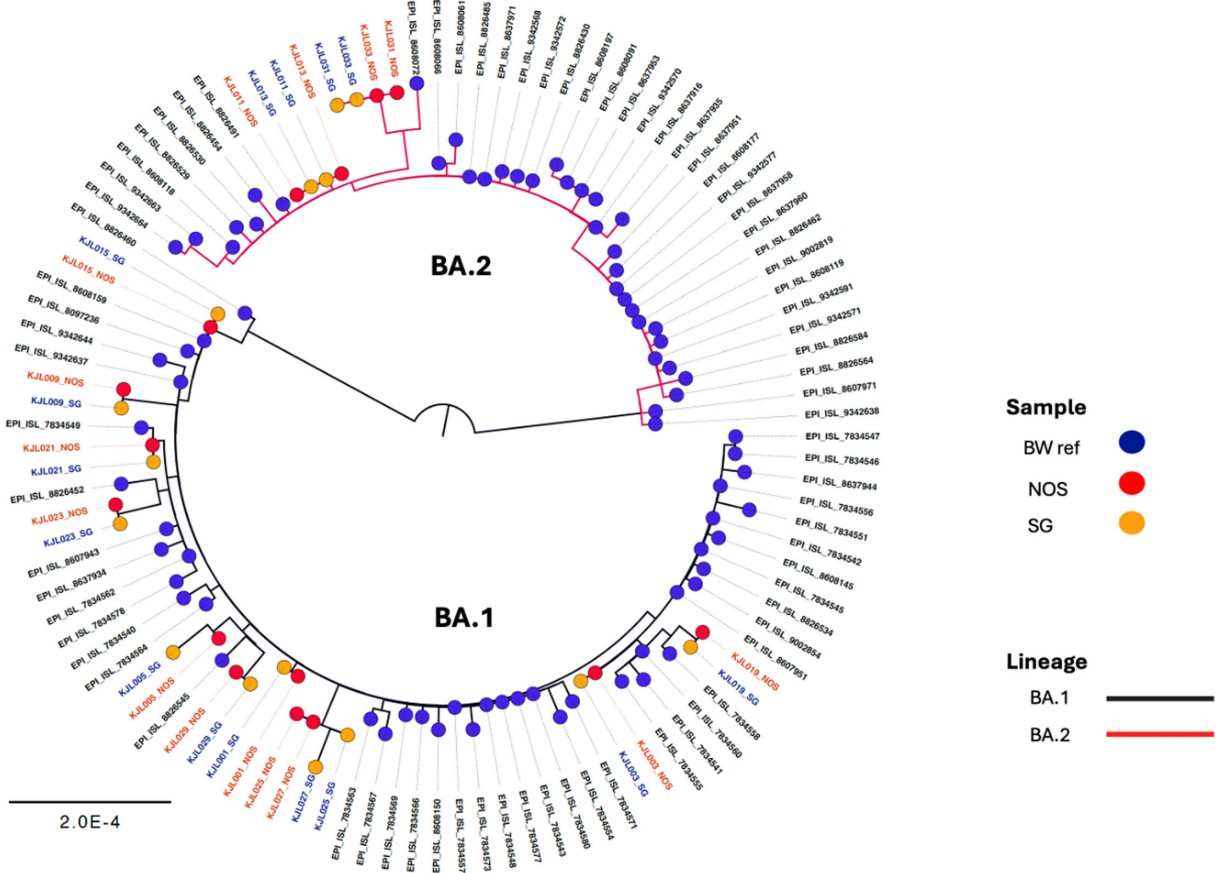

**FIG 1** Molecular phylogenetic analysis of NOS and SG samples by maximum likelihood method with circulating Botswana strains (BW) and their accession numbers as reference. BA.1 and BA.2 Omicron strains were detected in sequences.

offering self-collection of SG for routine testing, we could lower the risk of exposure for healthcare workers, reduce their workload, reduce the need for the acquisition of PPE, and decrease associated costs for these resources.

Ongoing SARS-CoV-2 sequencing has been vital in monitoring the emergence of new variants. Monitoring the variants of concern has been critical for public health response (24). The discovery of the Omicron variant, which the WHO designated a variant of concern, encouraged countries across the globe to strengthen their public health interventions and promote routine sequencing (25). In addition to the RT-PCR testing comparison of the SG samples to the NOS, we also investigated the comparison of the two sample types in NGS. Genome coverage was significantly higher in NOS sampling than in SG sampling. There were three outliers detected in the SG specimen which had low coverage and may have impacted the comparison ( Fig. S1). The overlap in IQRs, 90.0%–99.6% in NOS and 87.0%–99.2% in SG, suggests that the coverage distributions are similar to some extent. However, there was statistical significance that NOS coverage is higher. While the majority of the samples had the same branch lengths between the SG and NOS in the phylogenetic tree, we observed different branch lengths for samples KJL005 and KJL027. The branch lengths of the SG samples were shorter than the NOS branch length. For sample KJL005_SG, there were 10 positions in the genome that were not sequenced; and for KJL027_SG, 362 positions in the genome were not sequenced, and this may have caused the variation in branch length for the two samples. Overall, our results showed that the genomes generated from matched successful pairs were similar in mutations and at the nucleotide level. These findings further suggest that the SG may be a reliable option for SARS-CoV-2 detection and in the monitoring of variants for public health response through NGS.

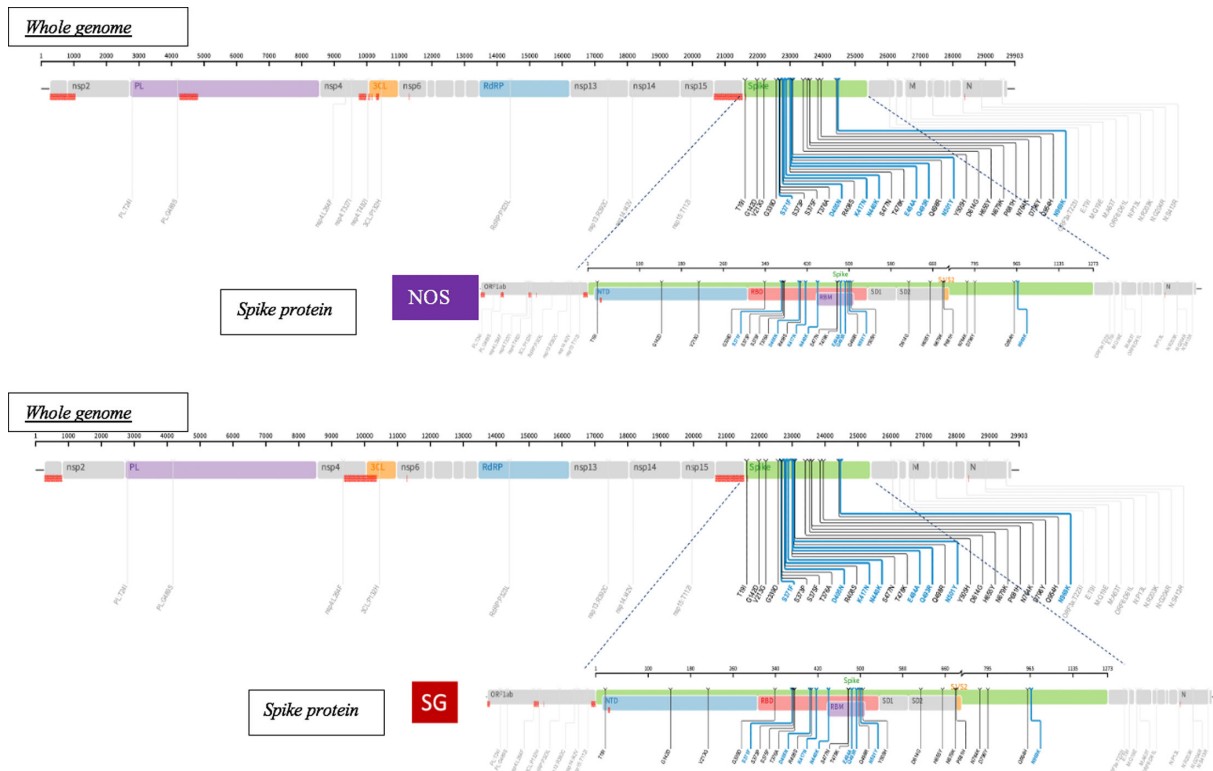

**FIG 2** Comparison of whole genomic structure (evenly distributed) and the spike protein between KJL011_NOS (EPI_ISL_15158766) and KJL011_SG (EPI_ISL_15158765) samples displaying the matched sequence mutations in the two sample types. The SG sampling method indicates that it is able to produce similar SARS-CoV-2 strains with similar mutations as the NOS sampling method.

## Limitations

While our study has highlighted the benefits of implementing the SG method for the detection and sequencing of SARS-CoV-2, there were weaknesses and research gaps observed. Due to stock-outs in reagents, there was inconsistency in PCR testing assays that may have affected the positivity detection rates in the samples due to differing thresholds across the different platforms. This may have overestimated or underestimated the calculated sensitivity rates. While the LineGene, ABI7500 FAST, and GeneXpert have all been validated for testing swabs and saliva samples for SARS-CoV-2 detection, they have not been validated for testing in saline gargle specimens. This may have contributed to the variation in the Ct values between NOS and SG samples. The reagent stock-out also resulted in the pausing of participant recruitment and introducing selection bias, where participants who were not approached for recruitment may have produced different findings compared to those who were recruited. The NOS sample collection was always collected before the SG sample. It is possible that this sampling order may have caused viral load to be reduced after NOS sampling, resulting in lower yield on the SG sampling. Perhaps a longer waiting period between the conduction of the two sampling methods may have translated to higher viral yield in SG sampling. Additionally, the efficacy of sampling methods needs to be further explored with controls of known SARS-CoV-2 viral concentrations. It is also possible that variation in viral loads may affect detection sensitivity in SG samples compared to NOS samples.

## Conclusion

Our clinical evaluation demonstrates the SG as a viable screening strategy, especially for wide-scale testing efforts in resource-limited settings. Although the Ct values of SG samples were higher than NOS, our study affirms the non-inferiority of the SG method in

detecting SARS-CoV-2 via qPCR and establishes that SG samples do not pose an obstacle for genomic sequencing of SARS-CoV-2. We found that the SG collection method had a relatively comparable sensitivity to NOS in the detection of SARS-CoV-2, and this correlates with observed findings from several similar studies. This method has the potential to overcome the challenges that come with swab-based sampling for SARS-CoV-2 testing and may be an alternative in testing for other viral pathogens. While the SG method may be a reliable alternative for SARS-CoV-2 detection and NGS, facilitating COVID-19 surveillance efforts in resource-constraint settings, additional studies are needed to address the feasibility and reliability of SG in a low- and middle-income setting for detection and sequencing of other respiratory viruses such as influenza and respiratory syncytial virus.

## MATERIALS AND METHODS

### Setting and study participants

The study was carried out with outpatients who presented to a COVID-19 testing facility in Gaborone, Botswana. Participants were eligible for the study if they were close contacts of a positive case and/or presented with clinical symptoms of COVID-19. Individuals under the age of 5 years old were excluded.

Eligible participants were invited to participate in the study and asked to provide an SG sample in addition to the routine HCW-collected NOS. Written consent was obtained from participants 18 years and above. Assent and parental consent were obtained from participants under the age of 18 years old. Demographic data including age, gender, and the date of symptom onset were also collected.

### Sample collection

The first case of COVID-19 in Botswana occurred on 30 March 2020, and by January 2021, there were over 20,000 cases (26), with numbers steadily rising. For our study, sample collection was performed over two periods. Participants were initially recruited from 24 December 2021 to 19 January 2022. Due to laboratory reagent and consumable stockout, recruitment was temporarily paused. It was later resumed, and participants were also recruited between 25 July 2022 and 16 September 2022. The method of sample collection was consistent over the two periods.

Participants were asked whether they had eaten or brushed their teeth 30 minutes prior to their recruitment into the study. Those who had not eaten or brushed their teeth within 30 minutes were eligible for participation. A flocked nasopharyngeal swab sample was collected from either the left or right naris based on the participant's preference, and a flocked oropharyngeal swab sample was collected from both tonsillar pillars and posterior oropharynx by a trained HCW. The swabs were both placed into a collection tube containing 3 mL viral transport media (BD Universal Viral Transport System). This procedure was then followed by SG collection. A single 5 mL vial of sterile 0.9% saline solution (Addipak, Teleflex Medical) was used for each participant's SG collection. The HCW squeezed the saline into the participant's mouth and instructed them to swish the contents for 5 seconds followed by tilting their head back and gargling for 5 seconds, and repeating this cycle for a total collection time of 20 seconds. The contents in the participant's mouth were then expelled into a sterile polypropylene collection container. Both matched SG and NOS samples were all collected under the observation of a HCW in the same manner and were stored under 4°C refrigerated conditions before being transported to the Botswana Harvard HIV Referral Laboratory (BHHRL) for routine same-day SARS-CoV-2 testing.

### Laboratory RT-PCR testing and sequencing

Paired SG and NOS samples collected between 24 December 2021 and 19 January 2022 underwent nucleic acid extraction using the high-throughput extraction machine, MGISP

960 platform (MGI Tech Co., Ltd., Shenzhen, China), followed by real-time polymerase chain reaction (PCR) using the DaAN Gene detection kit (DaAN Gene Co., Ltd., Guangzhou, China) on the ABI7500 FAST (Applied Biosystems, Thermo Fisher Scientific, Waltham, MA, USA) and LineGene (Bioer Technology Co., Ltd., Hangzhou, China) platforms. This assay detects the viral N and ORF1ab genes of the SARS-CoV-2 genome and the human RNase P gene present in the sample as control. Individual samples were considered positive in the analysis if there was a mean Ct value of less than 35 across the two viral targets. Paired SG and NOS samples collected between 25 July 2022 and 16 September 2022 were tested with the GeneXpert SARS-CoV-2 Assay (Cepheid, Sunnyvale, CA, USA). This assay detects the viral targets N2 and E genes and an internal sample processing control. Samples were considered positive as per the manufacturer's interpretation.

Both real-time PCR methods were validated by the BHHRL before the roll-out of nationwide COVID-19 testing in the facility. The real-time PCR results of both SG and NOS samples were available within 24 hours of collection. At the end of recruitment, lab-confirmed SARS-CoV-2-positive SG and NOS sample pairs (with mean Ct values less than 35) were retrieved from storage and underwent next-generation SARS-CoV-2 sequencing. Manual nucleic acid extractions were performed on both sample types using the DaAN gene extraction kit (DaAN Gene Co., Ltd., Guangzhou, China), followed by reverse transcription (RT)-PCR using LunaScript RT SuperMix kit (New England Biolabs, Ipswich, MA, USA) and TaqMan reagents (Thermo Fisher Scientific, Waltham, MA, USA) on the Bio-Rad platform (Bio-Rad Laboratories, Hercules, CA, USA). After library preparation, amplicons were pooled for nucleotide sequencing to be performed using the MinION platform (Oxford Nanopore Technologies, Oxford, UK) for NGS, following the manufacturer's instruction, the Nanopore Protocol version MRT_9127_v110_revH_14Jul2021 "PCR tiling of SARS-CoV-2 virus with rapid barcoding and Midnight RT PCR Expansion (SQK-RBK110.96 and EXP-MRT001)".

## Phylogenetic analysis

FastQ files were analyzed using Genome Detective (27) used to filter out poor quality reads, mapping to a reference SARS-CoV-2 genome Wuhan-Hu-1 (NC_045512.2) (28) and generating the BAM and consensus FASTA files for each sample. NextClade version 2.14.116 was used for the quality assessment of consensus sequences and mutational analysis. FASTA files were uploaded to this online tool to determine genome coverage of each sample and inferred Pango lineage from the nearest neighbor in the reference tree with 98% accuracy for recent sequences (29). This platform provided the SARS-CoV-2 variant detected in the uploaded sequences.

Two taxonomic assignment analyses were conducted as complementary methods to confirm one another. The IQ-tree software was used to generate a molecular phylogenetic tree with an implemented statistical maximum likelihood method (30). Phylogenetic Assignment of Named Global Outbreak Lineages software (31) was used to designate the sequences to known SARS-CoV-2 lineages. Sequences were deposited in the Global Initiative on Sharing All Influenza Data (GISAID) (32). The accession numbers have been provided in Supplementary document 2 as the EPI number after GISAID Indentifier.

## Statistical analysis

For the purposes of the analysis, a positive result on either sample type (SG or NOS) was used as a reference standard to calculate sensitivity. Sensitivity was reported with a 95% confidence interval. As we considered any positive result a true positive, we did not determine specificity. Sensitivity was compared between the SG and NOS sample types to analyze the comparability of testing using OpenEpi version 3.01 (33).

To determine median genome coverage, FASTA files of sequenced samples were uploaded onto NextClade version 2.14.116 and arranged from highest percentage genome coverage to lowest. The median value was identified for SG and NOS with

respective 25% and 75% interquartile ranges. A boxplot was designed to visualize the difference in coverage between the two sampling methods. The Wilcoxon rank-sum test was used to determine statistical significance ($P < 0.05$) using STATA v.18.5. Mean similarity at the nucleotide level in the sequence pairs was determined by assessing the pairwise distance in the aligned sequences using MEGA11 software (34). Genetic mutations in the sequence pairs were analyzed using the Stanford Coronavirus Antiviral & Resistance Database (35).

## ACKNOWLEDGMENTS

Many thanks to the study participants, the Block 8 Clinic in Gaborone staff, and the Botswana Harvard Health Partnership sequencing team.

This research study was funded by CICH-UBC (GR023778, PI: Dr. David Goldfarb). S.M., S.G., and W.T.C. were supported through the Sub-Saharan African Network for TB/HIV Research Excellence (SANTHE 2.0) through the Bill & Melinda Gates Foundation (INV-033558) and the National Institutes of Health NIH Fogarty International Center grants D43TW009610 and K43 TW012350. Sequencing was supported by the Bill & Melinda Gates Foundation (INV-036530), the National Institutes of Health Fogarty International Centre (3D43TW009610-09S1), and the HHS/NIH/National Institute of Allergy and Infectious Diseases (NIAID) (5K24AI131928; 5K24AI131924); the Africa CDC through the Pathogen Genomics Initiative; The Africa Pathogen Genomics Initiative (Africa PGI) at the Africa CDC is supported by the Bill & Melinda Gates Foundation (INV018978 and INV018278), Illumina Inc, the US Centers for Disease Control and Prevention (CDC), and Oxford Nanopore Technologies. The views expressed in this publication are those of the authors and do not necessarily represent the official positions of the funding agencies. The funders had no role in the study design, data collection and decision to publish, or in the preparation of the manuscript.

## AUTHOR AFFILIATIONS

[1]School of Allied Health Sciences, University of Botswana, Gaborone, Botswana
[2]Botswana Harvard Health Partnership, Gaborone, Botswana
[3]Botswana-UPenn Partnership, Gaborone, Botswana
[4]Department of Immunology & Infectious Diseases, Harvard T. H. Chan School of Public Health, Boston, Massachusetts, USA
[5]Division of Medical Virology, Department of Pathology, Stellenbosch University, Cape Town, South Africa
[6]University of British Columbia, Vancouver, Canada
[7]School of Medicine, University of Botswana, Gaborone, Botswana

## AUTHOR ORCIDs

Kwana Lechiile  http://orcid.org/0000-0001-6691-1905
Sikhulile Moyo  http://orcid.org/0000-0003-3821-4592
Wonderful T. Choga  http://orcid.org/0000-0001-7606-0569
Simani Gaseitsiwe  https://orcid.org/0000-0002-7089-3735
David M. Goldfarb  http://orcid.org/0000-0003-0835-9504

## FUNDING

| Funder | Grant(s) | Author(s) |
|---|---|---|
| University of British Columbia | GR023778 | David M. Goldfarb |
| Bill and Melinda Gates Foundation | INV018278 | Sikhulile Moyo |
| Bill and Melinda Gates Foundation | INV-033558 | Sikhulile Moyo |
| | | Wonderful T. Choga |

| Funder | Grant(s) | Author(s) |
|---|---|---|
| | | Simani Gaseitsiwe |
| National Institutes of Health | D43TW009610 | Sikhulile Moyo |
| | | Wonderful T. Choga |
| | | Simani Gaseitsiwe |
| National Institutes of Health | K43 TW012350 | Sikhulile Moyo |
| | | Wonderful T. Choga |
| | | Simani Gaseitsiwe |
| Bill and Melinda Gates Foundation | INV-036530 | Sikhulile Moyo |
| National Institutes of Health | 3D43TW009610-09S1 | Sikhulile Moyo |
| National Institute of Allergy and Infectious Diseases | 5K24AI131928 | Sikhulile Moyo |
| National Institute of Allergy and Infectious Diseases | 5K24AI131924 | Sikhulile Moyo |
| Bill and Melinda Gates Foundation | INV018978 | Sikhulile Moyo |

## AUTHOR CONTRIBUTIONS

Kwana Lechiile, Data curation, Formal analysis, Investigation, Methodology, Project administration, Supervision, Validation, Writing – original draft | Sikhulile Moyo, Formal analysis, Funding acquisition, Resources, Supervision, Writing – review and editing | Mai-Lei Woo Kinshella, Formal analysis, Project administration, Resources, Supervision, Writing – review and editing | Wonderful T. Choga, Formal analysis, Validation, Visualization, Writing – review and editing | Leabaneng Tawe, Supervision, Writing – review and editing | Jonathan Strysko, Project administration, Writing – review and editing | Gofaone Bagatiseng, Data curation | Iryna Kayda, Writing – review and editing | Kedumetse Seru, Methodology | Boitumelo J. L. Zuze, Methodology | Patience Motshosi, Methodology | Mosepele Mosepele, Conceptualization, Resources, Writing – review and editing | Irene Gobe, Supervision | Simani Gaseitsiwe, Methodology, Supervision, Writing – review and editing | Margaret Mokomane, Conceptualization, Resources, Supervision, Writing – review and editing | David M. Goldfarb, Conceptualization, Formal analysis, Funding acquisition, Investigation, Methodology, Project administration, Resources, Supervision, Writing – review and editing

## ETHICS APPROVAL

Through the collaborative work between two institutions, this project received ethical approval from the Health Research & Development Committee (HRDC) at Botswana's Ministry of Health (Protocol HRDC#00904) as Botswana is the site whichwhere the study was conducted, and the University of British Columbia (Protocol H21-01788) as the funders.

## ADDITIONAL FILES

The following material is available online.

### Supplemental Material

**Supplemental figure and tables (Spectrum02023-24-S0001.docx).** Figure S1 and Tables S1 to S3.
**Additional experimental details (Spectrum02023-24-S0002.pdf).** GISAID EPI_ISL IDs and the DOI for samples used to generate the phylogenetic tree.

Open Peer Review

**PEER REVIEW HISTORY (review-history.pdf).** An accounting of the reviewer comments and feedback.

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
