## [Reviewer comments · Microbiology Spectrum]

Microbiology Spectrum

Saline Gargle Collection Method is Comparable to Nasopharyngeal/Oropharyngeal Swabbing for the Molecular Detection and Sequencing of SARS-CoV-2 in Botswana

Kwana Lechiile, Sikhulile Moyo, Maggie Woo-Kinshella, Wonderful Choga, Leabaneng Tawe, Jonathan Stryko, Gofaone Bagatiseng, Iryna Kayda, Kedumetse Seru, Boitumelo Zuze, Patience Motshosi, Mosepele Mosepele, Irene Gobe, Simani Gaseitsiwe, Margaret Mokomane, and David Goldfarb

Corresponding Author(s): Kwana Lechiile, University of Botswana

Review Timeline:

Submission Date:	August 13, 2024
Editorial Decision:	January 28, 2025
Revision Received:	March 28, 2025
Accepted:	April 15, 2025

Editor: Po-Yu Liu

Reviewer(s): The reviewers have opted to remain anonymous.

Transaction Report:

DOI: <https://doi.org/10.1128/spectrum.02023-24>

Re: Spectrum02023-24 (Saline Gargle Collection Method is Comparable to Nasopharyngeal/Oropharyngeal Swabbing for the Molecular Detection and Sequencing of SARS-CoV-2 in Botswana)

Dear Ms. Kwana Lechiile:

Thank you for the privilege of reviewing your work. Below you will find my comments, instructions from the Spectrum editorial office, and the reviewer comments.

Revision Guidelines

Sincerely,
Po-Yu Liu
Editor
Microbiology Spectrum

Reviewer #2 (Public repository details (Required)):

NGS data has apparently been deposited but no accession numbers provided

Reviewer #2 (Comments for the Author):

Study summary

The study compared the saline gargle (SG) method to the traditional nasopharyngeal/oropharyngeal swab (NOS) method for SARS-CoV-2 detection and sequencing in Botswana. SG was found to be user-friendly, easier to collect, and highly sensitive, though slightly less so than NOS. Both methods apparently showed high accuracy in detecting the virus, with apparently similar sequencing results for the Omicron variant. The authors suggest SG as a viable alternative to NOS for respiratory pathogen testing, offering a simpler and less invasive approach.

The overall impression of the study

Saline gargle (SG) method is a promising and practical alternative to the traditional nasopharyngeal/oropharyngeal swab (NOS) method for SARS-CoV-2 detection. While NOS remains more sensitive, SG offers significant advantages in terms of ease of use, patient comfort, and accessibility, suggesting SG could be a reliable tool for viral detection, potentially simplifying large-scale testing for respiratory pathogens. The lower sensitivity however may reduce its reliability in detecting SARS-CoV-2, especially in low viral load cases. This manuscript requires major revision, a decision which is supported by comments/suggestions as follows:

Line 31: what is crossing threshold?

Table 1 and table 2: Will it be more appropriate to have the NOS as the reference method and SG as the testing method for the sensitivity analysis? then separate the results from RT-PCR and NGS into 2 separate table?

Figure 1: For sample KJL005 and KJK027, why the SG and NOS results have different branch length? Meanwhile the other sample sets have a similar branch length. Please explain. Additionally, please provide the scale bar for the Maximum likelihood phylogenetic tree in your figure

L116-120: Please provide the method used to calculate the median coverage? And coverage against what? Please list the genome coverage and sequencing depth for samples (with their EpiCoV accession number) that were subjected to NGS in a table format. Please explain the NGS methodology in methods.

L124: 80% coverage, do you mean genome coverage or sequencing coverage?

L126: Please provide details on how NextClade analysis were conducted in the method section.

Line 170: Is dispensing the swab in 3ml utm/vtm/saline not a routine practice? Could taking two specimens (NOS) prior to SG impacted on the low sensitivity observed with SG (viral load may have reduced as a result of NOS with less virus left behind for SG). Waiting at least 30 minutes between NOS and SG collection may have a positive impact? It is therefore important to use controls of known viral concentrations to check the efficacy of the methods.

Line 179: check references

Line 252: Please specify as to why ethical approval from two institutions was obtained.

Line 283: What was the Ct range of specimens used. What is the efficacy of SG on low viral load specimens. The study doesn't extensively discuss how varying viral loads might affect SG detection sensitivity compared to NOS. Testing dilutions of commercial viral preparations will provide further insights.

Please check line 294

Line 297: BHHRL-please write in full

Line 299: change small "v" to capital V in SARS-CoV-2

L312 and L330: which reference SARS-CoV-2 genome/s was/were use for the two analysis. Please include the NCBI accession number of these reference genomes?

L315: What is the reason behind for conducting two taxonomic assignment analysis (one with IQ-tree and another with PANGOLIN)?

L319: Please provide the assigned EpiCoV numbers by GISAID for your NGS sequences submitted to GISAID for verification of data.

L323: It is not clear how sensitivity was calculated. Why was specificity not calculated. What is the reference standard of a positive result? What program were use for the statistical analysis on sensitivity+specificity analysis? Please provide the sensitivity and specificity results along with the following information: sample size, Kappa (95% CI) and p-value from McNemar's test

L327: Please provide the citation reference for OpenEpi

L329: Where is the result for the Mean similarity comparison? if you are comparing similarities between nucleotides, then use average nucleotide identity (ANI), which is a measure of nucleotide-level genomic similarity between the coding regions of two genomes. MEGA is not the right software for this purpose.

L331+ Figure 2: for the Genetic mutations in the sequence pairs analysis, please provide details to the findings in this figure and how is that related to NOS and SG sample collection method? please also provide a table of the list of SNPs identified for each sample, and the comparison results in pairs (NOS vs SG per sample).

Supp Figure 1: Please add description of the plot and method.

Reviewer #3 (Comments for the Author):

This manuscript describes a comparison of nasopharyngeal/oropharyngeal swab (NOS) sampling versus saline gargle (SG) sampling for diagnostic SARS-CoV-2 molecular testing. Paired specimen collections were collected from individuals during the pandemic in two separate time frames utilizing two different molecular platforms (DaAN Gene detection kit/ABI7500 FAST and

Cepheid Gene Xpert assay), respectively. Next generation sequencing (NGS) was also performed in a subset of paired positive specimens which had SARS-CoV-2 CT values of <35 to assess specimen suitability for NGS testing. It is posed that SG collections are an appropriate alternative to NOS collections for COVID-19 testing which may be particularly important in resource-limited settings (collection ease, wider reagent availability, limited risk of complications, limited need for HCWs or HCW PPE, among others). The presented data support the utility of SG for testing as an alternative. Addressing the following comments will allow the reader to better assess the performance of SG collection for SARS-CoV-2 PCR and NGS testing performed at this institution.

Additional Comments:

- 1) Lines 92-99: It may be useful to create a separate result heading to discuss the patient specimens/demographics versus including this information under 'qPCR Testing'. It is recommended to break down the number of specimens collected by respective collection time frame (Dec. 2019-Jan. 2022 and July 2022-Sept. 2022) or by testing on respective molecular platform. Inclusion of other patient demographics like type of symptoms and time to symptom onset, etc., would also be useful to understand where patients were in their clinical course.
- 2) Table 2: As it is established that NOS specimens are 'gold standard' for testing, it seems unnecessary to retain Table 2 which shows the data with SG as the reference method.
- 3) Figure 2: It would be useful to include boxes or arrows to point out to the reader what is being compared. It may be worth renaming 'N/O' to 'NOS' to remain consistent with rest of manuscript.
- 4) Lines 163-164: Could the Ct value differences (SG vs. NOS) be described?
- 5) Line 166: What is the sensitivity of SG testing described in other papers, and what was the comparator in these studies?
- 6) Lines 177-178: Was patient favoring of SG over NOS something that was heard offhand, or were patient preferences documented in some way?
- 7) Lines 198-199: It is suggested from materials and methods that SG collections were obtained in an observed manner. Could it be confirmed if all SG collections were performed under HCW observation in the same manner?
- 8) The performance characteristics of the respective molecular platforms used for testing should be described in the limitations section, as well as what each molecular platform is officially validated for in terms of clinical specimens.

Microbiology Spectrum Point-by-Point Responses

Reviewer #2

Comment 1: NGS data has apparently been deposited but no accession numbers provided

Response 1: Accession numbers have now been provided as the ESPI_SET ID and the DOI.

Comment 2: What is the crossing threshold (line 31)

Response 2: Stated as <35 in line 31 “samples with mean crossing threshold <35”. For clarity, the sentence has been modified to “samples with mean crossing threshold value of <35”

Comment 3: Table 1 and table 2: Will it be more appropriate to have the NOS as the reference method and SG as the testing method for the sensitivity analysis? then separate the results from RT-PCR and NGS into 2 separate table?

Response 3: A reference standard of a positive result on either sample type (SG or NOS) was used to calculate sensitivity with the assumption that tests would not result in false positives. We felt this was justified as an approach particularly given that all subjects were presenting with for clinical testing for SARS-CoV-2 and that both samples were collected from the respiratory tract. Our group used the same approach for our prior publication in an ASM journal Goldfarb et al that was published in Journal of Clinical Microbiology, 2021 (<https://doi.org/10.1128/jcm.02427-20>). Also supporting this approach, this same reference standard was used for one of the largest systematic reviews comparing saliva samples to swab samples Bastos et al Annals of Internal Medicine (<https://doi.org/10.7326/M20-6569>).

Comment 4: Figure 1: For sample KJL005 and KJK027, why the SG and NOS results have different branch length? Meanwhile the other sample sets have a similar branch length. Please explain. Additionally, please provide the scale bar for the Maximum likelihood phylogenetic tree in your figure

Response 4: The difference in branch length is attributed to positions in the genome of the respective samples not being sequenced. This has been addressed in the discussion, Lines 240-245. A scale bar for the Maximum likelihood phylogenetic tree has been included in Figure 1.

Comment 5: L116-120: Please provide the method used to calculate the median coverage? And coverage against what? Please list the genome coverage and sequencing depth for samples (with their EpiCoV accession number) that were subjected to NGS in a table format. Please explain the NGS methodology in methods.

Response 5: Method used to calculate the median genome coverage for each sample type is included in the methods section (Lines 385-388). Coverage was mapped against a reference SARS-CoV-2 genome, NC_045512.2 (Line 363). A table has been included as Supplementary

Table 2. While exact depths for samples could not be recovered, we were able to express genome length and percent coverage in the samples. However, depth for samples is estimated to be 50X plus. The protocol used for NGS methodology has been stated in the methods (Lines 356-358).

Comment 6: L124: 80% coverage, do you mean genome coverage or sequencing coverage?

Response 6: Genome coverage. This has been amended to be more clear (now Line 136)

Comment 7: L126: Please provide details on how NextClade analysis were conducted in the method section.

Response 7: This has been included in the methods section (Lines 364-368, 386-389).

Comment 8: Line 170: Is dispensing the swab in 3ml utm/vtm/saline not a routine practice? Could taking two specimens (NOS) prior to SG impacted on the low sensitivity observed with SG (viral load may have reduced as a result of NOS with less virus left behind for SG). Waiting at least 30 minutes between NOS and SG collection may have a positive impact? It is therefore important to use controls of known viral concentrations to check the efficacy of the methods.

Response 8: We believe that one of the potential reasons the lower sensitivity observed in SG may be that higher volume (5ml) was used for collection compared to the 3ml utm. The larger volume in the SG method may have resulted in the dilution of the patient's sample prior to testing (Lines 196-200). The uncertainty of the impact in sampling order and the need to explore the efficacy of sampling methods with controls of known SARS-CoV-2 viral concentrations were elements that were initially overlooked and have now been added as study limitations (262-268).

Comment 9: Line 179: check references

Response 9: (now Line 207) Listed references have been cross-checked and are correct

Comment 10: Line 252: Please specify as to why ethical approval from two institutions was obtained.

Response 10: Ethical approval was sought from two institutions because of the collaborative work. Source of funding was from UBC which consists of its IRB, and the study was conducted in the Botswana setting through IRB approval from Botswana Ministry of Health. This has now specified in Lines 299-302.

Comment 11: Line 283: What was the Ct range of specimens used. What is the efficacy of SG on low viral load specimens. The study doesn't extensively discuss how varying viral loads might affect SG detection sensitivity compared to NOS. Testing dilutions of commercial viral preparations will provide further insights.

Response 11: (now Line 331) Samples were collected prospectively from symptomatic participants. It was unclear the Ct range of specimens at recruitment. The variation of VLs affecting SG detection sensitivity has now been highlighted as added limitation of our study (Lines 235-238).

Comment 12: Please check line 294

Response 12: Wording has been corrected. (now Line 342)

Comment 13: Line 297: BHHRL-please write in full

Response 13: BHHRL was written in full at prior line 278 (now Line 326) and abbreviation was continued going forward in the text (now at Line 346, previously 297).

Comment 14: Line 299: change small "v" to capital V in SARS-CoV-2

Response 14: This has been corrected

Comment 15: L312 and L330: which reference SARS-CoV-2 genome/s was/were use for the two analysis. Please include the NCBI accession number of these reference genomes?

Response 15: The reference genome was NCBI accession number NC_045512.2 and this has been included in the methods (Line 363).

Comment 16: L315: What is the reason behind for conducting two taxonomic assignment analysis (one with IQ-tree and another with PANGOLIN)?

Response 16: Two taxonomic assignment analyses were conducted as complementary methods to confirm one another. IQ-tree provided a phylogenetic confirmation of the clades and PANGOLIN provided a different algorithm of clade confirmation.

Comment 17: L319: Please provide the assigned EpiCoV numbers by GISAID for your NGS sequences submitted to GISAID for verification of data.

Response 17: These have now been provided and are included in the supplementary material.

Comment 18: L323: It is not clear how sensitivity was calculated. Why was specificity not calculated. What is the reference standard of a positive result? What program were use for the statistical analysis on sensitivity+specificity analysis? Please provide the sensitivity and specificity results along with the following information: sample size, Kappa (95% CI) and p-value from McNemar's test

Response 18: As mentioned in response #3 above, we considered any positive result a true positive, and therefore we could not determine specificity (Response 3). OpenEpi was used to perform the statistical analysis.

Comment 19: L327: Please provide the citation reference for OpenEpi

Response 19: This has been included in the references (Ref#33)

Comment 20: L329: Where is the result for the Mean similarity comparison? if you are comparing similarities between nucleotides, then use average nucleotide identity (ANI), which is a measure of nucleotide-level genomic similarity between the coding regions of two genomes. MEGA is not the right software for this purpose.

Response 20: Mean similarity result was indicated under results section (Line 142-133). MEGA software allows for computation of pairwise distances which helps determine similarity percentage between the two sample type sequences after they have been aligned.

Comment 21: L331+ Figure 2: for the Genetic mutations in the sequence pairs analysis, please provide details to the findings in this figure and how is that related to NOS and SG sample collection method? please also provide a table of the list of SNPs identified for each sample, and the comparison results in pairs (NOS vs SG per sample).

Response 21: The figure was meant to show that similar mutations were detected in the NOS and SG sampling – circulating Omicron strain at the time. This was further highlighting that SG is able to perform as well as NOS in genomic sequencing for SARS-CoV-2 . More details to the findings have been added in Lines 160-165, and in the description under Figure 2. A table of the list of SNPs has been included in the supplementary files (Supp Table 2).

Comment 22: Supp Figure 1: Please add description of the plot and method.

Response 22: This has been expanded (Supp Figure 1 description), including some modifications in the methods (Lines 389-392), results (Lines 128-131), and discussion (Lines 236-240).

Reviewer #3

Comment 1: Lines 92-99: It may be useful to create a separate result heading to discuss the patient specimens/demographics versus including this information under 'qPCR Testing'. It is recommended to break down the number of specimens collected by respective collection time frame (Dec. 2019-Jan. 2022 and July 2022-Sept. 2022) or by testing on respective molecular platform. Inclusion of other patient demographics like type of symptoms and time to symptom onset, etc., would also be useful to understand where patients were in their clinical course.

Response 1: A separate heading has been created and a summary table of patient demographics has been included in Results (Table 1). A more detailed table of each patient has been included as a supplementary table (Supp Table 1).

Comment 2: Table 2: As it is established that NOS specimens are 'gold standard' for testing, it seems unnecessary to retain Table 2 which shows the data with SG as the reference method

Response 2: The term “gold standard” is mentioned in the introduction and this comment has highlighted the confusion it brings for the remainder of the text. The term “gold standard” has now been removed from Line 91. A reference standard of a positive result on either sample type (SG or NOS) was used to calculate sensitivity with the assumption that tests would not result in false positives. Similar method approach was done in study by Goldfarb et al that was published in Journal of Clinical Microbiology, 2021. (<https://doi.org/10.1128/jcm.02427-20>)

Comment 3: Figure 2: It would be useful to include boxes or arrows to point out to the reader what is being compared. It may be worth renaming 'N/O' to 'NOS' to remain consistent with rest of manuscript.

Response 3: The figure has been modified to be and the description has been enhanced to make the figure clearer for the reader. Additionally, the Results section has been elaborated. (Lines 160-165)

Comment 4: Lines 163-164: Could the Ct value differences (SG vs. NOS) be described?

Response 4: This has been included in the discussion (Line 190)

Comment 5: Line 166: What is the sensitivity of SG testing described in other papers, and what was the comparator in these studies?

Response 5: This has been included in the discussion (Lines 187-189)

Comment 6: Lines 177-178: Was patient favoring of SG over NOS something that was heard offhand, or were patient preferences documented in some way?

Comment 6: This was something heard offhand from firsthand experiences (now Lines 204-206). There was no documentation.

Comment 7: Lines 198-199: It is suggested from materials and methods that SG collections were obtained in an observed manner. Could it be confirmed if all SG collections were performed under HCW observation in the same manner?

Comment 7: This has been confirmed in Lines 326.

Comment 8: The performance characteristics of the respective molecular platforms used for testing should be described in the limitations section, as well as what each molecular platform is officially validated for in terms of clinical specimens.

Response 8: This has been included in the limitations section Lines 257-260.

Re: Spectrum02023-24R1 (Saline Gargle Collection Method is Comparable to Nasopharyngeal/Oropharyngeal Swabbing for the Molecular Detection and Sequencing of SARS-CoV-2 in Botswana)

Dear Ms. Kwana Lechiile:

Your manuscript has been accepted, and I am forwarding it to the ASM production staff for publication. Your paper will first be checked to make sure all elements meet the technical requirements. ASM staff will contact you if anything needs to be revised before copyediting and production can begin. Otherwise, you will be notified when your proofs are ready to be viewed.

Sincerely,
Po-Yu Liu
Editor
Microbiology Spectrum

Reviewer #3 (Comments for the Author):

The reviewer comments have been addressed adequately. The use of "qPCR" is not recommended as the abbreviation to be used for the SARS-CoV-2 molecular methods that were used in this study. "qPCR" is generally interpreted as "quantitative polymerase chain reaction", however the methods used herein were not quantitative approaches, as in, no viral load was reported or can be ascertained from the methods used.

The following needs attention:

1. Use the term “cycle threshold” instead of crossing threshold
2. Figure 1 legend: include BW in brackets after Botswana strain
3. Close brackets after overall sensitivity of 96.9 (95% CI: 84.3-99.4 in abstract and in (SQK-RBK110.96 and EXP-MRT001 in methods
4. Line 362-363: please refine sentence
5. Line 374: please indicate that the accession numbers have been provided in supplementary table ... as the EPI number after GISAID
6. What is the reason behind for conducting two taxonomic assignment analysis (one with IQ-tree and another with PANGOLIN)? The response provided by the author to this comment should be included in the text in the manuscript
7. The sensitivity calculations need to be clearly explained in the manuscript. Was the reference standard used commercially purchased? etc